# Information nudges for influenza vaccination: Evidence from a large-scale cluster-randomized controlled trial in Finland

**Lauri Sääksvuori**[1,2,3]*, **Cornelia Betsch**[4,5], **Hanna Nohynek**[6], **Heini Salo**[6], **Jonas Sivelä**[6], **Robert Böhm**[7,8]

**1** Tampere University, Department of Health Sciences, Faculty of Social Sciences, Tampere, Finland, **2** University of Turku, INVEST Research Flagship Center, Turku, Finland, **3** Finnish Institute for Health and Welfare, Centre for Health and Social Economics, Helsinki, Finland, **4** University of Erfurt, Media and Communication Science and Center for Empirical Research in Economics and Behavioral Sciences, Erfurt, Germany, **5** Bernhard Nocht Institute for Tropical Medicine, Hamburg, Germany, **6** Finnish Institute for Health and Welfare, Infectious Disease Control and Vaccinations, Department of Health Security, Helsinki, Finland, **7** University of Vienna, Faculty of Psychology, Vienna, Austria, **8** University of Copenhagen, Department of Psychology and Copenhagen Center for Social Data Science (SODAS), Copenhagen, Denmark

\* lauri.saaksvuori@thl.fi

**Data Availability Statement:** This paper uses administrative health records maintained by the Finnish Institute for Health and Welfare. Access to

## Abstract

### Background

Vaccination is the most effective means of preventing the spread of infectious diseases. Despite the proven benefits of vaccination, vaccine hesitancy keeps many people from getting vaccinated.

### Methods and findings

We conducted a large-scale cluster randomized controlled trial in Finland to test the effectiveness of centralized written reminders (distributed via mail) on influenza vaccination coverage. The study included the entire older adult population (aged 65 years and above) in 2 culturally and geographically distinct regions with historically low (31.8%, $n = 7,398$, mean age 75.5 years) and high (57.7%, $n = 40,727$, mean age 74.0 years) influenza vaccination coverage. The study population was randomized into 3 treatments: (i) no reminder (only in the region with low vaccination coverage); (ii) an individual-benefits reminder, informing recipients about the individual benefits of vaccination; and (iii) an individual- and social-benefits reminder, informing recipients about the additional social benefits of vaccination in the form of herd immunity. There was no control treatment group in the region with high vaccination coverage as general reminders had been sent in previous years. The primary endpoint was a record of influenza vaccination in the Finnish National Vaccination Register during a 5-month follow-up period (from October 18, 2018 to March 18, 2019). Vaccination coverage after the intervention in the region with historically low coverage was 41.8% in the individual-benefits treatment, 38.9% in the individual- and social-benefits treatment and 34.0% in the control treatment group. Vaccination coverage after the intervention in the region with

health records is regulated in Finland under the Act on the Secondary Use of Health and Social Data (552/2019) and can be obtained by sending a direct request to the Finnish Institute for Health and Welfare (https://thl.fi/en). The authors are willing to assist in making data access requests. All statistical code used to organize and analyze the data is shared using the Open Science Framework and is permanently available at https://osf.io/qdrc4/ (DOI 10.17605/OSF.IO/QDRC4).

**Funding:** The authors received no external funding for this work. The costs of preparing (e.g. printing the letters and acquiring envelopes) and mailing the letters (postal fees) were paid by the Finnish Institute for Health and Welfare and the City of Espoo. The funders had no role in study design, data collection and analysis, decision to publish, or preparation of the manuscript.

**Competing interests:** I have read the journal's policy and the authors of this manuscript have the following competing interests: LS, HN, HS and JS declare grants to their employer, but no personal support or financial relationship, from Sanofi Pasteur and Innovative Medicines Initiative IMI during the conduct of the study. HN and HS declare membership in the National Advisory Committee on Vaccination in Finland. CB and RB declare no support from any organization or financial relationships with any organizations that might have an interest in the submitted work. All authors declare no other relationships or activities that could appear to have influenced the submitted work.

**Abbreviations:** COVID-19, Coronavirus Disease 2019; MDE, minimum detectable effect; PCV, pneumococcal conjugate vaccine; TBE, tick-borne encephalitis; TD, tetanus-diphtheria; WHO, World Health Organization.

historically high coverage was 59.0% in the individual-benefits treatment and 59.2% in the individual- and social-benefits treatment. The effect of receiving any type of reminder letter in comparison to control treatment group (no reminder) was 6.4 percentage points (95% CI: 3.6 to 9.1, $p < 0.001$). The effect of reminders was particularly large among individuals with no prior influenza vaccination (8.8 pp, 95% CI: 6.5 to 11.1, $p < 0.001$). There was a substantial positive effect (5.3 pp, 95% CI: 2.8 to 7.8, $p < 0.001$) among the most consistently unvaccinated individuals who had not received any type of vaccine during the 9 years prior to the study. There was no difference in influenza vaccination coverage between the individual-benefit reminder and the individual- and social-benefit reminder (region with low vaccination coverage: 2.9 pp, 95% CI: −0.4 to 6.1, $p = 0.087$, region with high vaccination coverage: 0.2 pp, 95% CI: −1.0 to 1.3, $p = 0.724$). Study limitations included potential contamination between the treatments due to information spillovers and the lack of control treatment group in the region with high vaccination coverage.

## Conclusions

In this study, we found that sending reminders was an effective and scalable intervention strategy to increase vaccination coverage in an older adult population with low vaccination coverage. Communicating the social benefits of vaccinations, in addition to individual benefits, did not enhance vaccination coverage. The effectiveness of letter reminders about the benefits of vaccination to improve influenza vaccination coverage may depend on the prior vaccination history of the population.

## Trial registration

AEA RCT registry AEARCTR-0003520 and ClinicalTrials.gov **NCT03748160**

## Author summary

### Why was this study done?

- Increasing levels of vaccine hesitancy threatens the progress made in halting vaccine-preventable diseases.

- There is an urgent need to evaluate the effectiveness of different behavioral interventions aiming to increase vaccination coverage.

- Pragmatic randomized controlled trials are critical for understanding how to increase vaccination coverage in real-world settings.

### What did the researchers do and find?

- This large-scale cluster-randomized controlled trial tested the effectiveness of centralized written reminders, distributed via regular mail, with various information contents on influenza vaccination coverage among the older adult population in Finland.

- This study showed that postal reminders are an effective and easily scalable intervention strategy to increase vaccination coverage.

- This study showed that the effectiveness of interventions aiming to improve vaccination coverage may depend on the prior vaccination history of the population.

- There was no difference between reminders that informed recipients about the individual benefits of vaccinations and reminders that informed recipients about the additional social benefits of vaccinations, such as herd immunity, in terms of their impact on influenza vaccination coverage.

### What do these findings mean?

- Reminder letters designed to address the psychological barriers that may prevent people from getting vaccinated effectively encourage vaccinations at close to zero marginal costs.

- Sending reminders to population groups with low vaccination coverage maximizes the effectiveness of reminder interventions.

## Introduction

Vaccinations have contributed enormously to global health. Large-scale vaccination programs continue to reduce morbidity and mortality due to numerous infectious diseases and comprise the backbone of health security strategies around the globe. However, increasing levels of vaccine hesitancy, defined as a delay in the acceptance or refusal of vaccines despite the availability of vaccination services, threatens the progress made in halting vaccine-preventable diseases [1–4]. In 2019, the World Health Organization (WHO) declared vaccine hesitancy to be one of the 10 biggest threats to global health. Understanding how to improve vaccine coverage and overcome different mechanisms underlying vaccine hesitancy is important, not only to improve current vaccination coverage but also to secure high coverage of new vaccines, such as the Coronavirus Disease 2019 (COVID-19) vaccines.

Several factors have been identified as relevant predictors of vaccine hesitancy. These factors include lack of trust in the safety and effectiveness of vaccinations (confidence); lack of appropriate disease-risk perception (complacency); perceived or actual structural and behavioral barriers, such as forgetting or difficulties in access (constraints); engagement in extensive information searching with potential risks of being exposed to misinformation (calculation); and lack of concern for vulnerable others (social responsibility) [4–6]. Despite accumulating evidence about the psychological antecedents of vaccination decisions and the development of validated measures to understand vaccine hesitancy, there is little population-based evidence about the effectiveness of scalable low-cost behavioral interventions that can be used to address specific factors associated with vaccine hesitancy.

Patient reminders and recall interventions via letters, email, or mobile phone messages are shown to be an effective method to increase vaccination coverage in outpatient, community-based, primary care settings [7,8]. Reminders address the psychological barriers that may prevent people from getting vaccinated (e.g., forgetting to make a vaccination appointment and

lack of practical information on how to make an appointment). Reminders can also communicate information about other factors related to vaccine hesitancy [9], such as the individual benefits of being vaccinated, which address complacency by providing information about disease risk.

Less attention has been paid to whether enhancing reminders with content that highlights the social benefits of vaccines could further increase vaccination coverage. Vaccinations not only incur individual benefits through direct protective effects but also affect the community at large through indirect effects, which reduce the risk of spreading the disease to others and build up herd immunity [10]. Highlighting these positive behavioral externalities could, in theory, increase the motivation for prosocial vaccination to protect unvaccinated individuals and lead to higher vaccination coverage. In fact, existing empirical research shows that educating individuals about the social benefits of vaccination can increase their social responsibility and vaccination intentions [11–13]. Consequently, sending reminders that provide information about the individual and social benefits of vaccinations could be a way to increase vaccination coverage.

We conducted a large-scale cluster-randomized controlled trial in Finland to test the effectiveness of centralized, one-time, written reminders (distributed via regular mail) that highlighted (i) the individual benefits of vaccinations or (ii) both the individual and social benefits of vaccination in increasing influenza vaccination coverage. The focus was on influenza vaccinations among the older adult population, where the gap between the vaccination target and actual coverage is particularly large [14]. After the intervention, data from comprehensive nationwide health records on influenza vaccination coverage were used to determine the effectiveness of the information treatments.

## Methods

### Study design

We conducted the trial in 2 geographically and culturally distinct communities in Finland. The trial had 2 active treatment arms. The first treatment highlighted the individual benefits of vaccination. The second treatment highlighted the individual and social benefits of vaccination. In the western region (Fig 1), there was a control treatment group without any intervention. In the southern region, there was no control treatment group because the local health authority had in previous years sent influenza vaccination reminder letters to the entire population aged 65 years and above. Thus, our intervention did not leave anyone without information that they would have otherwise received in the absence of the intervention. The study was conducted in partnership with local health authorities in both regions.

The treatments varied the information content of individual reminders (the original letters are available in S1 Appendix). The individual-benefits reminder contained basic information about the severity of influenza symptoms, seasonal influenza vaccination, the availability of vaccinations (locations and dates to receive the vaccine), and instructions about how to book an appointment with the vaccine administration. The individual- and social-benefits reminder provided the same information as the individual-benefits reminder but also contained the following information about herd immunity:

> "Your decision to vaccinate does not only protect you but others as well. Your vaccination may protect small children whose immune system is still developing. You will be able to protect your loved ones who are unable to get vaccinated. Your vaccination may prevent the spread of influenza viruses. Thus, the whole society benefits from your decision to vaccinate."

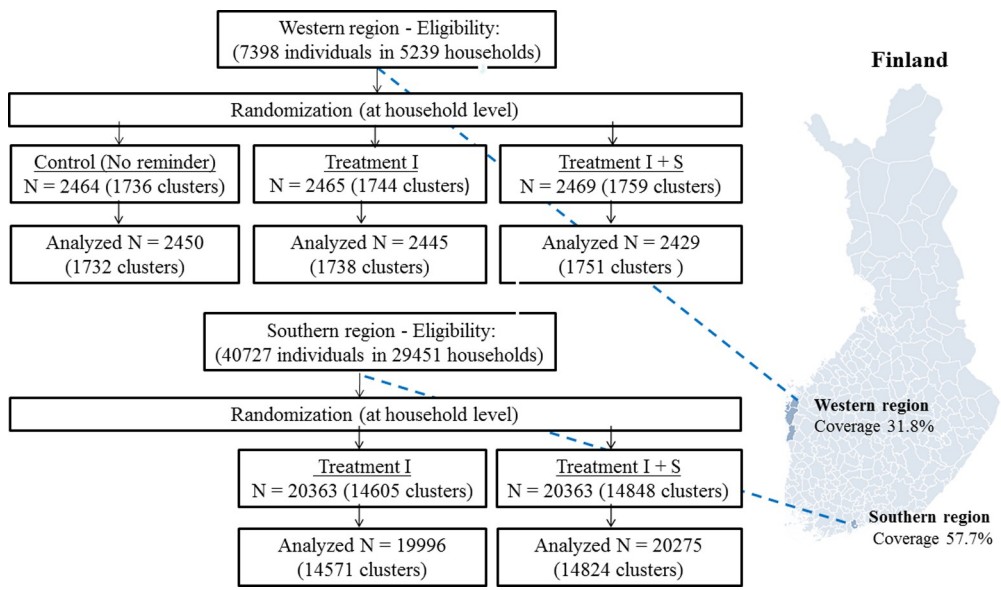

**Fig 1. Study regions and randomization scheme.** The map in Fig 1 was created for this article in R software using open source data (CC BY 4.0) from Statistics Finland. The base layer of the map used in Fig 1 is available at Statistics Finland's map service (https://tilastokeskus-kartta.swgis.fi/?lang=en). The R code and shapefiles to reproduce the map in Fig 1 are available at https://osf.io/v453z/. Control = no reminder, Treatment I = individual-benefit reminder, Treatment I + S = individual- and social-benefit reminder.

## Study population

The study took place in 2 regions with widely varying baseline vaccination coverage to test the effectiveness of different reminders in 2 different contexts and obtain information about the potential generalizability of findings across populations with differing baseline vaccination coverages and socioeconomic characteristics. The 2 regions represent populations with different socioeconomic characteristics and historical influenza vaccination coverage. The western region on the west coast of Finland is a rural region that contains 5 independent municipalities (Maalahti, Korsnäs, Närpiö, Kaskinen, and Kristiinankaupunki). The region has a single public provider of primary healthcare services that is co-owned by the municipalities. This region has low influenza vaccination coverage among people aged 65 years and older (31.8% during the influenza season of 2017 to 2018) compared to the national average (47.7% during the influenza season of 2017 to 2018). The southern region encompasses the city of Espoo, the second-largest city in Finland. The population in the southern region belongs to the inner urban core of the Helsinki metropolitan area and has one of the highest rates of influenza vaccination coverage among people aged 65 years and above (57.7% during the influenza season of 2017 to 2018).

The study population included everyone born in or before 1953 (aged 65 years and above) residing in the 2 regions on June 1, 2018. However, individuals living in housing units with more than 2 persons (e.g., nursing homes) were excluded from the sample and statistical analyses after the randomization because these units could include private nursing homes that provide seasonal influenza vaccinations to all residents as part of their care plan. Thus, the final analysis sample included only home-dwelling individuals living either in a single- or 2-person household (Fig 1). We excluded from the sample and statistical analysis also all individuals who received influenza vaccination before the beginning of the follow-up time. There were 10 individuals who received influence vaccination in the Western region between June 1, 2018 and October 17, 2018, and 210 individuals who received influenza vaccination in the Southern region between June 1, 2018 and October 17, 2018.

There were no other scientific, ethical, or economic reasons to exclude any individuals who met the specified inclusion criteria. Moreover, since the marginal costs of including additional individuals in these types of information interventions are very low, it was considered worthwhile to maximize the statistical power to detect even potentially small effect sizes. We focused on older adults aged 65 years and above, as they are entitled to free influenza vaccinations in Finland and identifiable from the population register by age. Older adults belong to a risk group with higher morbidity and mortality from influenza viruses than the prime working-age population [15,16].

## Randomization and masking

We used the Finnish Population Register to identify the name and postal address of individuals who met the eligibility criteria (age and place of residence). Randomization took place at the household (cluster) level to avoid sending reminders with different contents to the same household members. Randomization at the household level was implemented using unique apartment IDs and a computer-generated randomization code written by the authors (S1 Appendix). We used simple randomization without any blocking factors. The sample in the western region ($n =$ 7,398) was randomized into 3 treatment arms of equal size: (i) no reminder (control treatment group); (ii) individual-benefits reminder; and (iii) individual- and social-benefits reminder. The sample in the southern region ($N =$ 40,727) was randomized into 2 treatment arms of equal size: (i) individual-benefits reminder; and (ii) individual- and social-benefits reminder.

Individuals residing in the study regions and belonging to the target group were unaware of the study. The reminders themselves did not make any reference to any experimental variation in wording. Nurses administering influenza vaccinations during the follow-up period were not aware that different letters were sent to eligible individuals. We had no direct contact with either the recipients of the mailed letters or the nurses administering influenza vaccinations in the target region. We did not obtain informed consents for this study, because we did not recruit any participants and analyzed anonymous administrative data. The study protocol was approved by the Finnish Institute for Health and Welfare's Institutional Review Board (Decision Number: THL/1444/6.02.01/2018). The study protocol is available as a supporting information (S1 Protocol).

## Outcomes

The impact of different letters on vaccination coverage was measured at the individual level using administrative health records. The Finnish National Vaccination Register contains nationwide records of all vaccinations given at public healthcare units in Finland since 2009 [17]. We used lot numbers to identify vaccine types and time stamps to determine when they were administered. The main outcome variable was having (versus not having) received an influenza vaccination during a 5-month follow-up period (from October 18, 2018 to March 18, 2019). We also used prior vaccination history data to study the potential heterogeneity of the average treatment effects.

## Procedure

All reminders were sent via regular post to eligible individuals on October 17, 2018. All reminders were double-sided and written in both Finnish and Swedish to consider multilingual study populations. Individual identifiers (social security numbers) from the Finnish Population Register were used to match the received letters with complete vaccination records from the Finnish National Vaccination Register. The final dataset was produced using individual identifiers (encrypted social security numbers) that enabled us to merge population register data with administrative vaccination records. The final dataset did not contain any information that would allow for the direct identification of personal information.

## Trial registration

As this study spans multiple disciplines, we preregistered the experimental design and submitted the preanalysis plan to multiple registries: the US National Library of Medicine Registry for clinical trials (clinicaltrial.gov, trial number: 240317), the American Economic Association Registry for randomized controlled trials (trial number: AEARCTR-0003520), and aspredicted.org (trial number: #15682).

## Statistical analysis

Our randomized controlled trial included the entire population aged 65 years and above in the study regions. Consequently, we did not perform prospective sample size calculations. However, we report the minimum detectable effect (MDE) size for different treatment comparisons to assess whether potential null findings identify the absence of a true effect or signify a lack of statistical power. Our computations of MDE sizes do not account for potential corrections of multiple hypotheses testing. Taking into account the correlation of outcomes within (2-person) households, randomization at the household level, and a prior baseline vaccination rate of 31.8% in the western region, we computed that the (average) sample size of 2,441 individuals per treatment, divided into 1,740 clusters with an intracluster correlation of 0.7, was sufficient to obtain 80% power for a 5% (two-sided) level test for at least a 4.9 percentage point difference in the probability of receiving an influenza vaccination between any 2 treatments. Combining active treatment arms to estimate the impact of a reminder per se allows for the detection of smaller effect sizes with 80% power (Fig B in S1 Appendix).

The study population in the southern region was divided into 2 equally large treatment groups. Taking into account the prior baseline vaccination rate of 57.7% in the southern region, we computed that a sample size of 40,271 individuals, divided into 2 treatments and 29,395 clusters, was sufficient to obtain 80% power for a 5% (two-sided) level test for at least a 1.5 percentage point difference in the probability of receiving an influenza vaccination between the 2 treatments. More comprehensive power calculations that vary in statistical power and assumed intracluster correlations are available in S1 Appendix).

To determine the impact of reminders per se on influenza vaccination coverage, we estimated the pooled effect of the individual-benefit and the individual- and social-benefit treatments. We estimated statistical models using linear probability estimation, wherein the coefficient of the treatment indicator can be directly interpreted as the impact of the intervention on vaccination coverage. We used linear probability models for simplicity and ease of interpreting coefficient values. Table A in S1 Appendix provides results from logit models and multilevel mixed-effect linear models with an error structure that allows for cluster-level heterogeneity (random effects) at the household level. These alternative regression models provided extremely similar results. For reporting relative risk, we used a Poisson regression with standard errors clustered at the household level.

As preregistered, our primary statistical models did not include any control variables. However, we performed robustness analyses by running complementary linear probability models that controlled for prior vaccination histories and demographics (Table B in S1 Appendix). In addition, we assessed the robustness of our statistical estimates by running balance checks to test whether the random assignment successfully balanced demographics and individual vaccination histories across the treatment groups (Table 1).

To determine the impact of the different types of reminders on influenza vaccination coverage, we separately estimated the effects of the individual-benefits reminder and the individual- and social-benefits reminder. These models were estimated using linear probability models. In each model, we used standard errors clustered at the household level. We assessed the

robustness of our findings by estimating random effect models that included households as a random intercept (Table A in S1 Appendix). We adhered to the Consolidated Standards of Reporting Trials (CONSORT) checklist (S1 CONSORT Checklist) for conducting and reporting of this trial.

## Results

### Population and baseline characteristics

Table 1 displays the baseline characteristics across the regions and treatments, showing large differences in the proportion of previously vaccinated individuals between the western and southern regions. Influenza vaccination coverage was 31.8% in the western region and 57.7%

**Table 1. Summary statistics by study region and treatment (analysis sample).**

| | Descriptive statistics | | | Balancing tests—abs. standardized differences and *p*-values | | |
|---|---|---|---|---|---|---|
| *Panel A: Western region* | Control (N = 24,50) | Treatment I (N = 2,445) | Treatment I + S (N = 2,429) | I vs. Control | I + S vs. Control | I vs. I + S |
| Influenza vaccination, previous season | 787 [32.1%] | 818 [33.5%] | 724 [29.8%] | 0.028 (p = 0.401) | −0.050 (p = 0.139) | 0.078 (p = 0.020) |
| Influenza vaccination, any year | 1,097 [44.8%] | 1,113 [45.5%] | 1,033 [42.4%] | 0.015 (p = 0.656) | −0.045 (p = 0.178) | 0.060 (p = 0.070) |
| Any vaccination | 1,752 [71.5%] | 1,809 [74.0%] | 1,747 [71.9%] | 0.056 (p = 0.082) | 0.009 (p = 0.776) | 0.04 (p = 0.145) |
| Age | 75.6 (7.86) | 75.4 (7.79) | 75.3 (7.71) | 0.027 (p = 0.413) | −0.044 (p = 0.185) | 0.016 (p = 0.615) |
| Women | 1,268 [51.8%] | 1,270 [51.9%] | 1,256 [51.7%] | 0.004 (p = 0.842) | −0.001 (p = 0.961) | 0.005 (p = 0.808) |
| Single households | 1,011 [41.3%] | 1,027 [42.0%] | 1,072 [44.1%] | 0.015 (p = 0.659) | 0.058 (p = 0.090) | 0.043 (p = 0.209) |
| Joint test | | | | (p = 0.573) | (p = 0.218) | (p = 0.330) |
| *Panel B: Southern region* | Control | Treatment I (N = 19,996) | Treatment I + S (N = 20,275) | I vs. Control | I + S vs. Control | I vs. I + S |
| Influenza vaccination, previous season | - | 11,567 [57.8%] | 11,683 [57.6%] | - | - | 0.005 (p = 0.693) |
| Influenza vaccination, any year | - | 14,280 [71.4%] | 14,292 [70.5%] | - | - | 0.020 (p = 0.071) |
| Any vaccination | - | 16,243 [81.2%] | 16,380 [80.8%] | - | - | 0.011 (p = 0.304) |
| Age | - | 74.0 (6.91) | 73.9 (7.74) | - | - | 0.019 (p = 0.097) |
| Women | - | 11,398 [57.0%] | 11,573 [57.1%] | - | - | 0.002 (p = 0.816) |
| Single households | - | 9,145 [45.7%] | 9,372 [46.0%] | - | - | 0.010 (p = 0.414) |
| Joint test | | | | | | (p = 0.232) |

Note: This table summarizes descriptive characteristics at baseline by region and treatment, and reports results from balancing tests. Reported descriptive statistics are frequencies, except for the variable *Age*, which shows the average age by region and treatment. Square brackets report proportions (%) and parentheses show standard deviations. Three last columns show results from balancing tests. First row in each cell shows absolute standardized differences in covariates between treatments. Second row in each cell shows *p*-values based on linear regression models that cluster standard errors at household level. The joint test of orthogonality across all covariates is based on a regression that includes all available (6) covariates and tests the joint hypothesis that β1 = β2 = . . . β6 = 0. Control = no reminder, Treatment I = individual-benefit reminder, Treatment I + S = individual- and social-benefit reminder.

in the southern region at the end of the influenza season of 2017 to 2018. Notably, the differences in coverage were not limited to influenza vaccination. The proportion of individuals who had received any vaccination during the 9 years prior to the influenza season of 2018 to 2019 was 72.5% in the western region and 81.0% in the southern region. The average age in our samples was approximately 75 years. Most individuals were women and lived in households with 2 people 65 years and above.

Table 1 also shows results from balancing tests (absolute standardized differences and *p*-values). Using a critical statistical-significance threshold of $p < 0.05$, we find one statistically significant difference in covariate balance: Influenza vaccination coverage was higher in the previous influenza season in the individual-benefit treatment than in the individual- and social-benefit treatment in the Western region. This number of statistically significant imbalances is expected to arise by chance alone. To complete the baseline comparisons, we provide results from a joint test of significance across all 6 covariates and find that there are no systematic imbalances between the treatment arms at baseline.

## Confirmatory analyses (preregistered)

The primary analysis compared influenza vaccination coverage across the experimental arms in the western and southern regions. We report intention-to-treat results. Thus, individuals in all treatment arms were expected to remain in the initially assigned treatment group. The only potential sources of attrition were emigration or mortality after the postal address was extracted from the population register. There was no reason to expect attrition to be correlated with treatment.

We first report the proportions and differences in proportions of influenza vaccination coverage by treatment arm in the western and southern regions (Fig 2). The statistical analysis adjusts for clustering at the household level. In the western region, we observed the highest rate of vaccination coverage in the individual-benefits treatment (41.8%, 95% CI, 39.5% to 44.1%), the second highest rate in the individual- and social-benefits treatment (38.9%, 95%

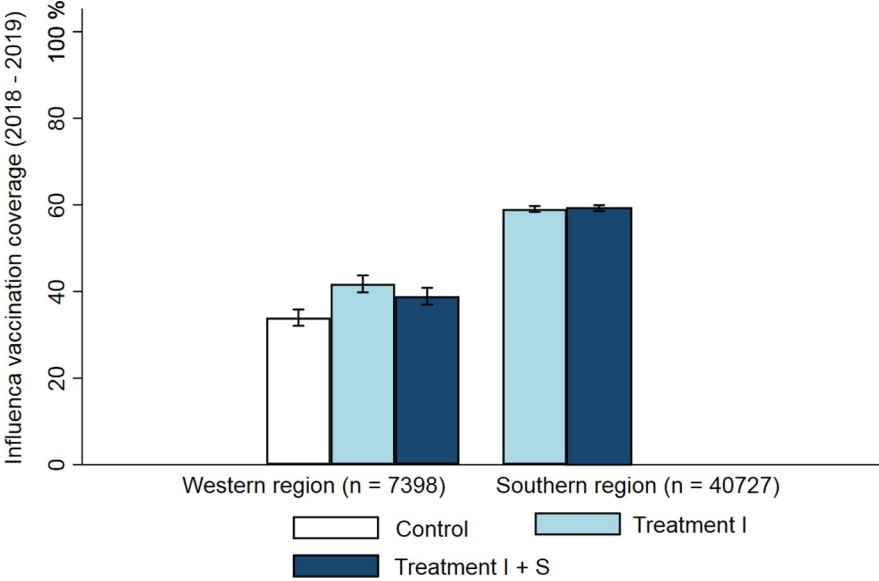

**Fig 2. Vaccination coverage by region and treatment.** Control = No reminder, Treatment I = individual-benefit reminder, and Treatment I + S = individual- and social-benefit reminder. Bar graphs denote influenza vaccination coverage. Error bars denote 95% confidence intervals.

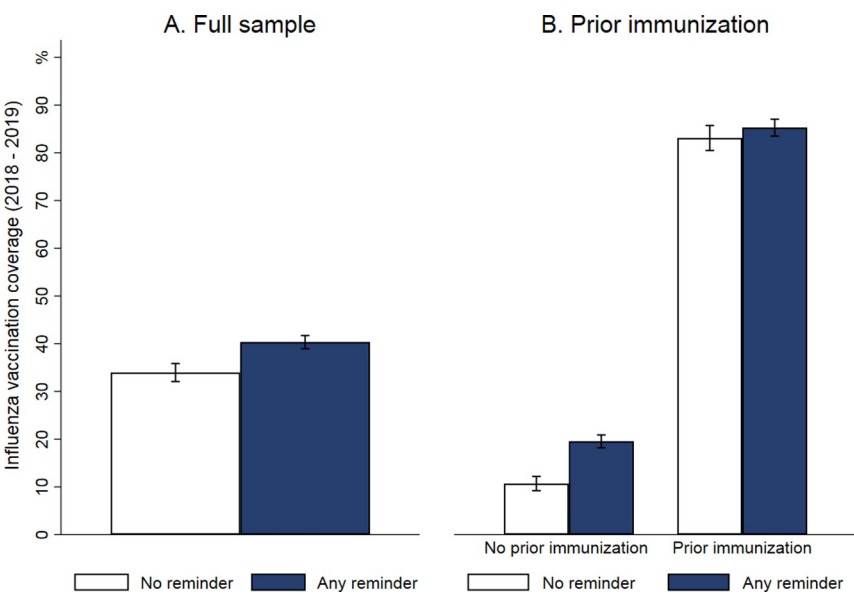

**Fig 3. Vaccination coverage by treatment in the western region.** Panel A: Full sample (No reminder vs. Any type of reminder, pooling the I and I + S treatments); Panel B: Vaccination coverage by treatment in the western region stratified by prior vaccination status (No reminder vs. Any type of reminder). Error bars denote 95% confidence intervals.

CI, 36.6% to 41.2%), and the lowest rate in the no reminder (control group) treatment (34.0%, 95% CI, 31.8% to 36.2%). The difference in proportions between the individual-benefits treatment and the control treatment group was 7.8 percentage points (95% CI: 4.6 pp to 11.0 pp, $p < 0.001$ | Risk ratio: 1.23 (1.13 to 1.34)), 4.9 percentage points (95% CI: 1.7 pp to 8.1 pp, $p = 0.002$ | Risk ratio: 1.15 (1.05 to 1.25)) between the individual- and social-benefits treatment and the control treatment group, and 2.9 percentage points (95% CI: −0.4 pp to 6.1 pp, $p = 0.087$ | Risk ratio: 1.07 (0.99 to 1.16) between the individual-benefit treatment and the individual- and social-benefit treatment. Finally, we pooled both reminder treatments (Fig 3A) and found that the effect of receiving any type of reminder versus being in the control treatment group without a reminder was 6.4 percentage points (95% CI: 3.6 pp to 9.1 pp, $p < 0.001$ | Risk ratio: 1.19 (1.10 to 1.28)).

In the southern region, we observed that vaccination coverage was similar in the individual- and social-benefit treatment (59.2%, 95% CI, 58.5% to 60.0%) and in the individual-benefit treatment (59.0%, 95 CI, 58.3% to 59.8%). Consequently, the difference in proportions of vaccination coverage between the individual-benefit treatment and individual- and social-benefit treatment was small (0.2 percentage points, 95% CI: −1.0% to 1.3%, $p = 0.724$ | Risk ratio: 1.00 (0.98 to 1.02)), indicating that there was no difference in vaccination coverage between the 2 reminder treatments.

## Exploratory analyses (not preregistered)

We explored the effect of reminders conditional on prior vaccination history. Moreover, we estimated possible cross-vaccination spillovers from influenza vaccinations to other common vaccinations among the age group. Only data from the western region were used in the analyses, as only this subdesign included a group of individuals who did not receive either reminder.

We estimated the treatment effect of reminder letters conditional on one of 3 indicators of individual vaccination history: having versus not having received an influenza vaccination

**Table 2. The effect of written information letters on influenza vaccination coverage conditional on prior vaccination history in a region with historically low vaccination coverage (Western region).**

| | Influenza vaccination coverage (Western region) | | | | | |
|---|---|---|---|---|---|---|
| | Conditional on influenza vaccination 2017–2018 | | Conditional on influenza vaccination 2011–2018 | | Conditional on any vaccination 2011–2018 | |
| | (1) Vac. | (2) Unvac. | (3) Vac. | (4) Unvac. | (5) Vac. | (6) Unvac. |
| Regression Coef.: Effect of any reminder (vs. no reminder) | 0.019 (.017) | 0.088*** (.012) | 0.048** (.019) | 0.084*** (.011) | 0.050*** (.017) | 0.053*** (.013) |
| Risk ratio: Effect of any reminder (vs. no reminder) | 1.021 (.018) | 1.824*** (.163) | 1.071** (.030) | 2.340*** (.312) | 1.131*** (.041) | 2.058*** (.417) |
| Observations | 2196 | 5128 | 3243 | 4081 | 5308 | 2016 |
| Coverage in control group (%) | 87.3% | 10.7% | 68.0% | 6.3% | 45.5% | 5.0% |

Notes: Reported regression coefficients are estimated using linear probability models. Reported risk ratio coefficients are estimated using Poisson regression. All models are estimated at the individual level. Standard errors in parentheses are clustered at the household level. Indicators for prior vaccination in Models 1 and 2: having vs. not having received influenza immunization during the previous seasonal influenza period (2017–2018); in Models 3 and 4: having vs. not having received any influenza immunization during the 9 years (2009–2018) prior to the influenza season of 2018–2019; Models 5 and 6: having vs. not having received any immunization during the 9 years (2009–2018) prior to the influenza season of 2018–2019.

*** $p < 0.01$

** $p < 0.05$, * $p < 0.1$.

during the previous seasonal influenza period (2017 to 2018); having versus not having received an influenza vaccination during the 9 years prior to the influenza season of 2018 to 2019 (from 2009 to 2018); and having versus not having received any vaccination during the 9 years prior to the influenza season of 2018 to 2019 (from 2009 to 2018). The length of the prior vaccination period (9 years) was based on data availability and maximized the available length of individual vaccination histories before the treatment assignment.

Table 2 (columns 1 and 2) shows the joint effect of any type of reminder versus no reminder conditional on influenza vaccination status during the influenza season of 2017 to 2018 (1 year prior to the study). We found that the effect of receiving any type of (versus no) reminder on vaccination coverage was substantially larger among previously unvaccinated individuals (8.8 percentage points higher in the reminder versus no reminder conditions, which corresponded to a relative increase of 82%) than among previously vaccinated individuals (1.9 percentage point increase).

Table 2 (columns 3 and 4) shows the joint effect of any type of reminder versus no reminder conditional on having versus not having received an influenza vaccination during the previous 9 years. We found that receiving versus not receiving a reminder increased vaccination coverage by 8.4 percentage points (relative increase of 134%) among individuals who had not received an influenza vaccination during the previous 9 years. For those who had received at least 1 influenza vaccination during the past 9 years, the increase was 4.8 percentage points (relative increase of 7%).

Table 2 (columns 5 and 6) shows the effects of receiving versus not receiving a reminder conditional on having versus not having received any type of vaccination during the previous 9 years. We found a substantial positive effect (5.3 percentage points) even among the most consistently unvaccinated individuals. As overall influenza vaccination coverage in this unvaccinated group was low (5.0%), the relative effect size of receiving any reminder was very large among the most consistently unvaccinated individuals (106%).

Finally, we examined whether receiving a reminder about the importance of influenza vaccinations increased vaccination coverage for other common vaccinations among the study population. These analyses utilized the fact that our data included comprehensive patient

records of all vaccinations received after the implementation of the intervention. We estimated cross-vaccination spillovers separately for the 3 most common types of vaccinations (other than influenza vaccinations), in this age-group: the pneumococcal conjugate vaccine (PCV), the tetanus-diphtheria (TD) vaccine, and the tick-borne encephalitis (TBE) vaccine. Moreover, we estimated the effect of receiving a reminder on the receipt of any other vaccine than influenza vaccine. Our results are reported in Table D in S1 Appendix and strongly indicate that there were no cross-vaccination spillovers. The estimated effects in all models were bounded to a tight interval around zero.

## Discussion

The aim of this study was to investigate the effect of 2 different types of centralized written reminders (distributed via regular mail) on influenza vaccination coverage among the older adult population. We observed that a low-cost and scalable intervention relying on individually mailed reminders substantially increased influenza vaccination coverage in a population with low baseline vaccination coverage. However, our results suggest that there was no difference in influenza vaccination coverage between the individual-benefits reminder and the individual- and social-benefits reminder in either the region with historically low influenza vaccination coverage or the region with historically high influenza vaccination coverage.

Comprehensive patient records enabled us to measure the effect of reminders conditional on individuals' prior vaccination history. The analyses revealed that the effect of reminders was substantially larger among individuals who had not received an influenza vaccination in the previous year. We also observed that even the most consistently unvaccinated individuals, who had not received any vaccination during the previous 9 years, responded positively to written reminders. By contrast, there was no statistically significant effect among previously vaccinated individuals. These findings suggest that the cost-effectiveness of interventions aiming to improve vaccination coverage may depend on the prior vaccination history of the target population.

Our results suggest that a written explanation of the social benefits of vaccinations, in addition to individual benefits, did not increase influenza vaccination coverage. In other words, we found that appealing to social responsibility, in addition to decreasing complacency, did not affect influenza vaccination coverage in our study population. Consequently, we conclude that, at least in the context of influenza vaccination and the reminder intervention used, communicating the social benefits of vaccination in the form of herd immunity leads neither to prosocial vaccination nor free riding on the vaccination efforts of other community members.

Our paper extends the study of behavioral interventions from hypothetical vaccination intentions and small-scale outpatient settings to a large-scale cluster-randomized controlled trial in which vaccination decisions are measured using comprehensive health records that include information about all vaccinations received before and during the follow-up period. The use of data from administrative health records had several key advantages. First, we were not restricted to studying vaccination intentions or self-reported vaccination outcomes but were able to objectively measure whether and when a vaccination occurred. Second, individuals residing in the study regions were not aware that different reminders were sent to eligible individuals. As a result, the generalizability of our results is not limited by the common concern that experimental results based on voluntary participation do not generalize to a population that was not aware of the experiment or that did not volunteer for the experiment when offered the opportunity. Third, the use of data from administrative health records enabled a sample size an order of magnitude larger than in typical randomized controlled trials that require the use of survey instruments to measure outcome variables. Finally, administrative

health records of all vaccinations enabled us to measure potential behavioral spillovers to other age-appropriate vaccinations. More generally, this study serves as an example of how a randomized study design can be merged with high-quality administrative data to estimate causal effects in large and representative samples. Using comprehensive and exact administrative information about prior vaccination histories, or statistical variables that predict prior vaccination history in the absence of exact health records, constitutes a promising way to enhance the effectiveness of behavioral interventions aiming to improve vaccination coverage.

Our findings are largely consistent with the literature that has documented the effectiveness of patient reminders and recall interventions on vaccination coverage [7,8,18–20]. However, most of the existing evidence stems from outpatient provider office settings in which there is an active care relationship between the provider and patient. The conclusions from these studies may not necessarily apply to large-scale interventions within the general older adult population. In contrast, our study overcomes these limitations and tests the effectiveness of centralized reminders as an easily scalable and low-cost communication strategy in the general older adult population.

This paper is related to nascent literature that has tested the effectiveness of various communication strategies and behavioral interventions on vaccination coverage across different vaccinations and populations [21–24]. There is increasing evidence that communicating the social benefit of herd immunity using short texts or images without sufficiently explaining the underlying mechanisms (e.g., using interactive simulations [11]) is ineffective at increasing vaccination intentions [25–27]. Hence, the observed null result may partly be due to an ineffective communication format but could also relate to the well-known intention–behavior gap [28]. However, it remains to be studied whether communicating the social benefits of vaccination in the form of herd immunity increases vaccine uptake against more contagious diseases with more exact threshold for herd immunity, such as measles. Overall, our results parallel findings from the literature, which indicate that information materials tailored using behavioral science techniques have, at best, only a modest effect on vaccination coverage. It may also be that behavioral interventions motivate those who plan to vaccinate but does not persuade vaccine-hesitant individuals [29,30]. In contrast, there is some evidence from low- and high-income countries that modest in-kind incentives and direct monetary incentives may increase vaccination coverage [31–33].

We acknowledge that our study has several limitations. First, there could have been some contamination between the treatments if information about the reminders and their contents were shared between individuals (e.g., neighbors, friends, and other individuals in the receiver's social networks) who belonged to different treatment groups. However, these kinds of information spillovers were minimized by the cluster-randomized design, which guaranteed that the same information would be received by all members of the same household. Second, the effectiveness of reminders may be underestimated, as we report intention-to-treat effects that disregard questions about the effectiveness of reminders among individuals who opened and read the letters. While the postal service in Finland is generally efficient and reliable, we could not obtain information about the proportion of letters that were successfully delivered, opened, and read by the recipients. The fact that the letters were written as centralized reminders (with printed letterheads and signatures by the local chief physicians) in collaboration with the Finnish Institute for Health and Welfare likely minimized recipients' concerns about their authenticity. Third, we were unable to identify the impacts of reminders per se on influenza vaccination coverage in the southern region, because all the individuals in this region were assigned to either the individual-benefits treatment or the individual- and social-benefits treatment. Thus, we are not able to infer whether the effect of receiving any reminder depends on the aggregate rate of vaccination coverage in the study population.

In conclusion, this large-scale cluster-randomized controlled trial has shown how a behavioral intervention study can be combined with routinely collected high-quality administrative data to estimate causal effects in large and representative samples. We observed that a reminder informing older adults about the benefits of vaccination led to a substantial increase in influenza vaccination coverage in a population with low baseline vaccination coverage. This positive effect on influenza vaccination coverage was observed even among the most consistently unvaccinated individuals. These findings have meaningful implications for the financing of preventive health interventions and public health authorities that implement vaccination communication strategies to enhance vaccine uptake and curb the spread of infectious diseases.

## Supporting information

**S1 CONSORT Checklist. CONSORT Checklist.** CONSORT, Consolidated Standards of Reporting Trials.
(DOC)

**S1 Appendix. Table A.** Average treatment effects estimated using linear probability models, logit models, and generalized mixed effects regression with random effects. **Table B**. Average treatment effects with and without control variables. **Table C.** The effect of reminders on influenza vaccine coverage by prior immunization history—Random effects linear model. **Table D.** Cross-vaccination spillovers to other age-appropriate vaccines. **Fig A.** Minimal detectable effect sizes (with $\alpha = 0.05$ and 0.80) for treatment comparisons by intracluster correlation coefficients in the Western region. **Fig B.** Minimal detectable effect sizes (with $\alpha = 0.05$ and 0.80) for the joint effect of any type of reminder by intracluster correlation coefficients in the Western region. **Fig C.** Minimal detectable effect sizes (with $\alpha = 0.05$ and 0.80) for the treatment comparison by intracluster correlation coefficients in the Southern region.
(DOCX)

**S1 Protocol. The original protocol/research plan.**
(DOCX)

## Author Contributions

**Conceptualization:** Lauri Sääksvuori, Cornelia Betsch, Hanna Nohynek, Heini Salo, Jonas Sivelä, Robert Böhm.

**Data curation:** Lauri Sääksvuori.

**Formal analysis:** Lauri Sääksvuori.

**Investigation:** Lauri Sääksvuori, Cornelia Betsch, Hanna Nohynek, Heini Salo, Jonas Sivelä, Robert Böhm.

**Methodology:** Lauri Sääksvuori, Cornelia Betsch, Robert Böhm.

**Project administration:** Lauri Sääksvuori, Jonas Sivelä.

**Visualization:** Heini Salo.

**Writing – original draft:** Lauri Sääksvuori, Cornelia Betsch, Hanna Nohynek, Heini Salo, Jonas Sivelä, Robert Böhm.

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
