## [Editor Report · Decision Letter 0]

11 Jun 2020

Dear Dr Sääksvuori, 

Thank you for submitting your manuscript entitled "Information nudges for influenza vaccination: Evidence from a large-scale cluster-randomized controlled trial" for consideration by PLOS Medicine.

Your manuscript has now been evaluated by the PLOS Medicine editorial staff and I am writing to let you know that we would like to send your submission out for external assessment.

Kind regards,

Richard Turner, PhD

Senior editor, PLOS Medicine

rturner@plos.org

---

## [Decision Letter · Decision Letter 1]

27 May 2021

Dear Dr. Sääksvuori,

Thank you very much for submitting your manuscript "Information nudges for influenza vaccination: Evidence from a large-scale cluster-randomized controlled trial" (PMEDICINE-D-20-02666R1) for consideration at PLOS Medicine. We do apologize for the long delay in sending you a response. 

Your paper was discussed among the editors and sent to independent reviewers, including a statistical reviewer. The reviews are appended at the bottom of this email and any accompanying reviewer attachments can be seen via the link below:

[LINK]

In light of these reviews, we will not be able to accept the manuscript for publication in the journal in its current form, but we would like to invite you to submit a revised version that addresses the reviewers' and editors' comments fully. You will appreciate that we cannot make a decision about publication until we have seen the revised manuscript and your response, and we expect to seek re-review by one or more of the reviewers. 

We hope to receive your revised manuscript by Jun 17 2021 11:59PM. Please email us (plosmedicine@plos.org) if you have any questions or concerns.

Please let me know if you have any questions, and we look forward to receiving your revised manuscript. 

Sincerely,

Richard Turner, PhD

rturner@plos.org

It seems that your trial was registered at clinical trials.gov on November 20, 2018. Please explain how this relates to the expectation that trials are registered prospectively. 

Please quote the study dates in your abstract. 

Please quote aggregate demographic details for study participants in the abstract. 

In the abstract and throughout the paper, please quote p values alongside 95% CI, where available. 

Please combine the "Methods" and "Findings" subsections of the abstract. 

The final sentence of the new combined subsection should begin "Study limitations include ..." or similar and should quote 2-3 of the study's main limitations. 

Please remove the information about funding from the abstract (in the event of publication, this information will appear in the article metadata, via entries in the submission form). 

After the abstract, we will need to ask you to add a new and accessible "Author summary" section in non-identical prose. You may find it helpful to consult one or two recent research papers published in PLOS Medicine to get a sense of the preferred style. 

The final paragraph of the Introduction would seem more appropriate in the Discussion section, and we ask you either to remove or reposition it within the text.

Please restructure the early part of your Discussion section so that the first paragraph provides a summary of the study findings.

Please use the general style "aged 65 years".

Throughout the text, please style reference call-outs as follows: "... externalities [6,7]." (noting the absence of spaces within the square brackets). 

In the reference list, please convert italics into plain text. 

Author lists should be formatted as follows (reference 1): Salmon DA, Dudley MZ, Glanz JM, Omer SB.

Noting reference 25, please ensure that all references have full access details. 

Again, please remove the information on study funding and competing interests from the end of the main text, as this should appear only in the article metadata.

Please supply a completed CONSORT checklist as a supplementary file, labelled "S1_CONSORT_Checklist" or similar and referred to as such in the Methods section. 

In the checklist, please refer to individual items by section (e.g., "Methods") and paragraph numbers, not line or page numbers as these generally change in the event of publication. 

Please attach the study protocol as a supplementary file, referred to in the Methods section.

Comments from the reviewers:

*** Reviewer #1: 

This is a statistical review of manuscript PMEDICINE-D-20-02666R1. The manuscript is well written and the topic is particularly important nowadays given the current Sars-Cov-2 epidemic. I have a few important comments that require clarification. 

Comments:

* Alignment between statistical section and Appendix C: appendix C uses 30% vaccination rate while the text uses 32%. So the text reports a MDE of 3.5% and the appendix 3.7%. The appendix uses 2450 participants per group, but the text uses 2465. I think that it is important to clarify in the text that you're effectively calculating the MDE. In other words it's not a prospective sample size calculation. 

* "We estimated also these statistical models using linear probability estimation where the

coefficients represent marginal effects." Please clarify, what do you mean by marginal effects in this instance ? 

* Discrepancy between text and abstract: "the effect of receiving any written information letter versus being in the control group without any written information was 6·4 percentage points (95% CI: 3·6 pp to 9·1 pp, p < 0·001)". The CI in the abstract is 4.1 to 8.8 pp. Please verify that all results match between the abstract and the text.

* Table A1: "To take into account the clustered randomization design linear probability models and logit models use standard errors clustered at household level, random effect model includes household as a random intercept". However, in the text, I find "In all regression models, we used standard errors that are clustered at the household level". I think that the text needs to be amended. 

Minor comments:

* "the probability of receiving an influenza vaccination between any two treatments". It is actually the between the control group and one of the active arms (either I, or I+S).

* Table A1: p=0.000 does not exist. Please replace by P < 0.000

*** Reviewer #2: 

This is a paper describing a cluster randomized trial evaluating the impact of a tailored information letter (about social benefits) vs a standard information letter vs no-letter on improving influenza vaccinations of older adults >65yrs of age in Finland. There appears to be two trials going on simultaneously: in the southern region of Finland (where baseline influenza vaccination rates are around 57%) it was a 2-arm comparative effectiveness trial testing standard letter message vs standard plus social benefit message within a letter. In the Western region (where baseline influenza vaccination rates are 32%), it was a 3-arm trial with control vs standard letter vs standard plus social benefit message in the letter. 

The study found no benefit in the southern region suggesting no impact of social benefit messaging, but a benefit of either message over no-letter control in the western region, suggesting some benefit of any letter in a population with very low baseline rates. The study also found larger effect if no prior vaccination (in the western region with very low rates). I believe the intervention involved a single letter, and the clustering involved families. 

Study strengths are that the question of testing the added value of a message with social benefit is a good one, and the system that captures vaccinations in Finland is comprehensive and also includes prior vaccinations so this is a true population-based study. It is also nice to see RCTs.

The study has a number of limitations:

Overall

The overall writing is highly complex, difficult to follow, often repeats, and is convoluted. I needed to read it several times to understand the paper. The methods and overall study design are fine, rather it is the writing that makes this paper challenging to follow

Introduction

The introduction of the paper states that the paper is designed to understand behavioral determinants of individual vaccination decisions—yet this study does not do that except that it compares whether adding societal benefit message helps. That is not the same as "understanding behavioral determinants. 

The introduction seems to wander back and forth across topics. It does not cite the major Cochrane review of reminder/recall in 2018, but instead cites a JAMA paper (that was linked with a Cochrane review) in 2000. Throughout, it seems naïve about the literature on reminders for vaccinations. This type of reminder is called "centralized reminders" by immunization experts since the reminders were sent from a central group rather than physician practices. 

Methods

While the methods are generally fine, the description of the methods is very complex and convoluted. I could not figure out the # letter reminders sent (I assume one), 

Results- 

The analyses seem written in a complex manner yet they are actually quite simple. Table 1 is confusing in that it is unclear why language was assessed. Table 2 is confusing. The figures are clear. 

Discussion 

The discussion section again says that the study assesses behavioral determinants of individual vaccination decisions - but it is not clear to this reviewer how it does that. I do see that it tested one behavioral intervention (social benefit). I believe the authors are trying to refer to behavioral economics research. On the other hand, the discussion fails to discuss the issue of very low baseline rates in the western region which may account for the bigger effects.

*** Reviewer #3: 

The authors present the results of an interesting pragmatic trial of the effect of letters, including standard information or information about herd immunity/social emphasis) versus no letter across two geographic regions in Finland. A particular strength of this study is the careful selection of two different regions with different prior vaccination coverage to enhance generalizability and explore the effect of letters among patients with different prior exposure to the vaccine. Another strength is the long duration of individuals included in the study (>9 years) and complete coverage of influenza vaccination outcomes through population-level claims. The results largely confirm a large prior trial conducted in the US (along with other trials of mailed letters) that show that mailed letters can increase vaccination rates and that additional tailoring does not provide massive additional better compared with no letter. 

Major comments

- Tailored letter: The authors emphasize that the letter is "tailored" about social benefits of vaccination due to herd effect. I posit instead that the letter emphasizes social consequences rather than just informational; it is not exactly tailored to individuals, so the language is inaccurate. It is also inconsistent, as the letters are described differently in the abstract versus the body of the manuscript. The authors should stick with what is written in the manuscript, as that appears more accurate.

- No control group in one region: The authors did not have a control group in the southern region (the region with better vaccination uptake). This means that the control group is likely biased towards being worse than the intervention group already (and therefore makes the overall results less believeable), because the only controls were in the region where there was worse baseline vaccination coverage. The authors recognize this as a limitation but need to be more careful in their interpretation, especially against control. It is understandable why no letter was sent, but the authors need to present results more clearly (including in the abstract) for just the Western region separately to overcome this. 

Minor comments

- Abstract (Findings): The authors state that individually mailed letters increased influenza vaccination coverage, but it is not clear which arm (treatment) the results refer to, nor the comparison. This should be clarified. The results for all 3 arms should be presented, rather than subgroups, which is what is being presented now. The results across arms is shown in the last sentence, but the results should be presented independently. 

- Statistical analysis (Power): The authors state that they assume an ICC of 0.5; is there citation for this or baseline data to support?

- Table 1: The authors should provide some data (e.g., absolute standardized differences or p-values) to describe balance in the groups. There is some evidence of imbalance across the arms on for example influenza vaccination receipt in prior season within region across arms, which biases towards seeing an effect in the letters (e.g., 33.5% vs. 32.1%). 

- Control variables: The authors state that they did not include any control variables in the analysis but should adjust for imbalanced covariates in at least supplementary analyses. 

- Figure 2: The scale should be 0-100% to avoid overinterpretation. 

Discretionary comments

- Abstract (Methods): The authors do not provide information about the modeling or statistical analysis plan in the abstract.

***

[LINK]

---

## [Decision Letter · Decision Letter 2]

3 Nov 2021

Dear Dr. Sääksvuori,

Thank you very much for submitting your revised manuscript "Information nudges for influenza vaccination: Evidence from a large-scale cluster-randomized controlled trial" (PMEDICINE-D-20-02666R2) for consideration at PLOS Medicine. 

Your paper was discussed among the editors and seen again by two of our reviewers, and by a new statistical reviewer. The reviews are appended at the bottom of this email and any accompanying reviewer attachments can be seen via the link below:

[LINK]

In light of these reviews, we will not be able to accept the manuscript for publication in the journal, but would like to invite you to submit a further revised version that addresses the reviewers' and editors' comments fully. We will be unable to make a decision about publication until we have seen the revised manuscript and your response, and we may seek re-review by one or more of the reviewers. 

We hope to receive your revised manuscript by Nov 23 2021 11:59PM. Please email us (plosmedicine@plos.org) if you have any questions or concerns.

Please let me know if you have any questions, and we look forward to receiving your revised manuscript. 

Sincerely,

Richard Turner, PhD

Senior editor, PLOS Medicine

rturner@plos.org

Please add "in Finland" or similar to the title. 

We suggest adding a few words to the abstract to briefly explain the reason for the difference between the two regions in terms of the control condition. 

Please quote the study's primary endpoint early in the "Methods and findings" subsection of your abstract.

Please adapt the "Conclusions" subsection of the abstract to begin "In this study, we found that ... was an effective ..." and adapt the tense of the remaining text in this subsection to match. 

Please bullet the individual points in the "Author Summary".

At the start of the "Discussion" section (Main text) we suggest "The aim of this study ...", and make that "... two types of written reminder".

In the reference list, please use the journal name abbreviations "PLoS ONE", "BMJ" and "JAMA", and abbreviate other journal names as appropriate.

Noting reference 6 and others, please list only 6 author names, followed by "et al.".

Noting reference 7, please ensure that all references have full access information. 

Please add "U S A" to reference 27.

Comments from the reviewers:

*** Reviewer #2: 

This is a revised paper describing a population-level cluster randomized trial evaluating the impact of a tailored information letter (about social benefits) vs a standard information letter vs no-letter on improving influenza vaccinations of older adults >65yrs of age in Finland. There were two trials going on simultaneously: in the southern region of Finland (where baseline influenza vaccination rates are around 57%) it was a 2-arm comparative effectiveness trial testing standard letter message vs standard plus social benefit message within a letter. In the Western region (where baseline influenza vaccination rates are 32%), it was a 3-arm trial with control vs standard letter vs standard plus social benefit message in the letter. The study found no benefit in the southern region suggesting no impact of social benefit messaging, but a benefit of either message over no-letter control in the western region, suggesting some benefit of any letter in a population with very low baseline rates. The study also found larger effect if no prior vaccination (in the western region with very low rates). Perhaps the two largest added values of this paper are: (1) population-level study, and (2) is the sub-analysis by prior vaccination, showing that in this setting the intervention(s) had greater impact among those who did not receive prior influenza vaccinations, although the level of vaccine receipt rose to only about 20% in that subgroup.

The initial submission received generally positive reviews with some critiques, which the authors have addressed. Some of the critiques included unclear writing in numerous spots, which the authors have largely addressed. I have some remining minor critiques:

Abstract:

The Conclusion states this is a "low-cost" intervention yet the abstract does not describe costs, so this should be deleted. The last statement may be an exaggeration since the results shown include findings for both the standard letter plus the behavioral letter (and no difference between standard and behavioral letters) overall. I would suggest writing: "the effectiveness of letter reminders about the benefits of vaccination to improve influenza vaccination coverage may depend on the prior vaccination history of the population." I also did a "caveat" since other studies have not found this finding. 

What do these findings mean?

I am confused by the mention of "text-based reminders"—do the authors have another study that involved text message reminders?

Introduction

This is now improved

Methods

I clicked on supplemental materials and I see (in English) the Individual benefit letter but cannot see the social good letter (I + S) that was used in one of the regions. So it is hard for me to judge whether the lack of a difference between the two letters was because including social good is not helpful in this region or whether the wording was not optimal in the social good letter. 

Results

The text and figures/tables are mostly fine.

The Cochrane reviews have relied on relative risks (unadjusted and adjusted) and not just percents and regression analyses (with regression coefficients as in Table 2). It would be helpful to display relative risks, without lengthening the already very long Results section. 

For Table 2—the subheadings could be better (under 1,2,3,4,5,6) to show what each columns mean (rather than only having that in the footnotes)

As a comment, the literacy level of the individual benefit letter is quite high. It is peculiar that it would have a greater effect than the individual benefit plus social good letter. 

Discussion

This section is now good. 

One suggestion I have is to also list some large-scale studies that have NOT found an effect of reminders. It turns out that increasingly, simple influenza vaccine reminders such as what was used here, at least in the US, have been found to not be effective in a variety of settings. 

*** Reviewer #3:

 Major comments:

1) Abstract: The abstract does not provide the referent group for the 6.4pp increase; is this compared with control or another type of letter? The authors state that individually mailed letters increased influenza vaccination coverage, but it is not clear which arm (treatment) the results refer to, nor the comparison. This should be clarified. The results for all 3 arms should be presented, rather than subgroups, which is what is being presented now. The results across arms is shown in the last sentence, but the results should be presented independently. In addition, some aspects of the abstract use words that could be over-interpreted, such as "crucially depends on", which is not supported by the data presented.

Minor comments:

2) Randomization: Was 1:1 simple cluster randomization implemented or did it account for any blocking factors? This is not currently clear. 

3) Outcome capture: Is there any chance that individuals could have gotten the flu shot outside of the 5-month follow-up period (particularly before)? How were these individuals treated in the analysis? It would likely bias towards the null but would be important to clarify. 

4) Households: Were any households more than 2-person? (e.g., siblings living with married siblings) The manuscript assumes only 2 adult persons/household. 

5) Sample size calculations: It is not clear if the authors adjusted for multiple testing in the analysis (e.g., through Bonferroni). This is likely ok given their study question (and the null result) but would be good to clearly specify. 

6) Table 1: The authors have not provided a description of balance changes across characteristics (e.g., ASDs or p-values) - this should be in Table 1, not online appendices. This was not sufficiently addressed in the prior revision.

7) Table 2: Please indicate which region these analyses are conducted in in the table. Also please include full titles of information to reduce the need for reviewers to go to footnotes.

8) Abstract: "lack of no reminder treatment": clearer to say "no control"

Discretionary comments:

9) "Text-based reminders" sounds like "text messaging" - recommend re-writing.

10) It is confusing to abbreviate to "Treatment I" and "Treatment I+S" - recommend putting more in a technical appendix but not to use non-standard abbreviations

*** Reviewer #4: 

The previous statistical reviewer was not available for the revision. I have checked the changes made in addressing reviewers comments and am satisfied the authors have addressed the statistical comments well.

***

[LINK]

---

## [Decision Letter · Decision Letter 3]

22 Dec 2021

Dear Dr. Sääksvuori,

Thank you very much for re-submitting your manuscript "Information nudges for influenza vaccination: Evidence from a large-scale cluster-randomized controlled trial in Finland" (PMEDICINE-D-20-02666R3) for consideration at PLOS Medicine.

I have discussed the paper with editorial colleagues and it was also seen again by one reviewer. I am pleased to tell you that, provided the remaining editorial and production issues are fully dealt with, we expect to be able to accept the paper for publication in the journal.

[LINK]

We hope to receive your revised manuscript in the first week of January. Please email us (plosmedicine@plos.org) if you have any questions or concerns.

Please let me know if you have any questions, and we look forward to receiving the revised manuscript.   

Sincerely,

Richard Turner, PhD

rturner@plos.org

Requests from Editors:

In the data statement (submission form) please finalize the statement (noting the current temporary wording "... will be shared ... after acceptance", noting PLOS' data policy, https://journals.plos.org/plosmedicine/s/data-availability.

In the "Methods and findings" subsection of the abstract, please adapt the sentence beginning "There was no control treatment in the region with high vaccination ..." to note that the control treatment was not included as general reminders had been sent in previous years. 

In the abstract, please use the style "7398" and "40,727".

Please use the phrasing "... in the control treatment group" or "... with the control treatment", for example, in the abstract and throughout. 

In the final subsection of the abstract, please amend the text to "In this study, we found that ... was an effective ...".

In the reference list, noting reference 3 and others, please use the conventional journal name abbreviations, e.g., "N Engl J Med." and "Proc Natl Acad Sci U S A.".

Noting reference 6, please ensure that all citations have full access details. 

Please spell out the journal or source name for reference 30.

Comments from Reviewers:

*** Reviewer #3: 

The authors have been sufficiently responsive.

***

[LINK]

---

## [Editor Report · Decision Letter 4]

18 Jan 2022

Dear Dr Sääksvuori, 

On behalf of my colleagues and the Academic Editor, Dr Lauffenburger, I am pleased to inform you that we have agreed to publish your manuscript "Information nudges for influenza vaccination: Evidence from a large-scale cluster-randomized controlled trial in Finland" (PMEDICINE-D-20-02666R4) in PLOS Medicine.

Prior to final acceptance, please quote the trial registration number (e.g., at clinical trials.gov) on the title/abstract page. 

PRESS

Sincerely, 

Richard Turner, PhD 

rturner@plos.org